`images/logo.png`

# Immunopeptidome Cluster Profiling
# with Critical Variable Selection

## I ABSTRACT

**Despite the worldwide effort to gain a deeper comprehension of the molecular mechanism that characterizes cancer, its blueprint is still elusive.**

**In the last decade, results in the context of cancer immunology helped researchers to identify a set of biological objects, namely antigens, which now represent a promising direction. Mass spectrometry is the typical investigation involved in collecting molecules.**

**This method and modern techniques allow a limited but satisfactory throughout on peptide knowledge.**

**However, the diversity coming out from such experiments is still poorly characterized.**

**In this paper, *UnChAnTies* (Unsupervised characterization of antigen communities) is introduced.**

**This tool is a computational platform that enables unsupervised clusterings and identifies critical variables characterizing each community of antigens. The goal of the pipeline is to identify relevant structures through unsupervised feature extraction with a model-free approach. Additionally, shed light, that takes part in the interplay between various kinds of cancer through their antigen are determined.**

## II INTRODUCTION AND MOTIVATION

In the last years, machine learning had an essential impact on immunology and cancer studies. The use of this technology allows determining more effective applications on the tumor treatments.

In addition, most of the neoantigens predicted computationally fail to elicit an immune response in vivo [1], meaning that design principles of antigens are poorly understood and the epitope - the terminal to which T-cells receptors bins- prediction is still limited.

Cancer can be characterized by the alterations heap in the genome and the typical cell regulatory activity alteration. These facts have long been known to result in antigen proliferation, resulting in peptides bound to primary histocompatibility class I (HMC-I) molecules on cancer cells' surfaces, distinguishing them from their regular counterparts.

The eradication of all its cells is required to prevent cancer proliferation. During the last two decades, the possibility to tackle cancer employing the capability of immunity emerged and is currently known as *Cancer Immunology*. This research field introduced novel both experimental and computational techniques (eg, respectively mass-spectrometry immunopeptidomics [2], [3] and machine learning-based methods [4] [5]).

The adaptive immune system is the largest source of genetic diversity inside the human body. A crucial locus for the immune system, located in chromosome 6, *Human Leukocyte Antigen* (HLA) contributes to more than half of the four to five million single-nucleotide polymorphisms (SNPs) in each genome. This locus encodes cell-surface proteins responsible for the regulation of the immune system.

Since pioneering the work of Boon [6], we know that $DC8^+$ T cells, spontaneously produced by patients, can recognize such cancer-specific peptide-MHCI complexes, making the detection of cancer cells feasible for the immune system.

On the physiological side, the immune system response to cancer is based on the *Cancer-Immunity Cycle* (CIC) [7]. Ideally, during this process, the T-Cells should recognize, reach and destroy cancer cells. The CIC is articulated in 7 main steps:

- Cancer cells spontaneously dies releasing antigens in the tissue environment (1): such antigens are captured by dendritic cells and presented to the T-cells (2)

- The T receptor cells do not recognize such antigens as a native from the organism (histocompatibility is broken) and activate (3)

- Starting from a lymph node and through blood vessel the T cells trafficking occurs until the cancer bed is reached (4)

- T cells infiltrate in tumor (5), recognize (6), and kill (7) cancer cells.

When the T-cells reach and kill cancer cells, antigens are newly released, and the process restarts. Thus, the CIC works as a positive feedback loop if optimally performed until the antigens are not released anymore.

However, in cancer patients, the CIC is disrupted and suboptimally performed. If one of the steps mentioned above is compro-

mised, the cycle is no longer possible, and the cancer population proliferates.

This fact allows cancer to select its antigens (surface peptides) repertory thanks to its continuous mutations and escape the immune surveillance as a consequence of the interaction with the immune system. These escape mechanisms are implemented by selecting MHC class I-deficient tumor escape variants.

On the other side, we have cancer antigens, i.e., ensembles of molecules with which the T-Cells recognize cancer as a pathogen. We call such ensemble the *immunopeptidome*. Overall the immunopeptidome, only a few objects can be regarded as a candidate for being included in immunotherapy. This particular quality, known as tumoral specificity, gives rise to peptides that arise from the expression of somatically mutated genes, namely, the Neoantigens. In the language of statistical physics, Neoantigens are points in a high-dimensional space corresponding where a gain function is optimized by cancer cells selection. Understanding the structure of the gain function landscape can help to decode the design principle of cancer evolution as a reverse engineering problem. Some clinical trials already employed neoantigen successfully [8], but why this happens remains elusive. However, these partial results stimulated the scientific community to discover the cancer-specific antigens mediating immune-system response or, in general, to enhance positive feedback in the CIC.

Inspired by the vast literature regarding reverse engineering of proteins [9] which is primarily based on the *Potts Model*, we were interested in a model-free description of biological objects. Such description actually exists and is known in the statistical physics community as *critical variable selection* (CVS) [10], which can be considered, in modern machine learning terms, an unsupervised feature extraction algorithm. This approach is promising, as it was shown to capture statistical moments beyond the second one and find long-range structures (eg. beyond linear motives) in the context of protein families.

The algorithm was introduced to find an easily interpretable explanation to the regularities found in multiple sequence alignment (MSA) of protein families. In general, evolution con is thought of as a process towards optimization of some biological function. If such optimization exists, there should be relevant sites, organized as (eventually non-contiguous) subsequences occurring with a broad distribution in the MSA. This special kind of occurrence was shown to encode both evolutionary and structural properties in the multiple sequence alignment of proteins. The described tools have been primarily employed in the context of multiple sequence alignment (MSA) of protein families [11]. We can regard a protein family as a cluster of evolutionary-related clusters of sequences.

Our main idea is to employ this unsupervised, model-free method to multiple sequence alignment of peptides, to find the relevant variables that characterize them. Unfortunately, immunopeptidomics data are not organized as protein families are. We thus created our peptide multiple sequence alignment (pepMSA) after clustering a custom dataset of epitopes. After this clustering procedure, we profiled each cluster with the *CVS* algorithm, mapping the diversity of the immunopeptidome landscape, finding a rich, formerly unobserved, structure that will help to design better therapies in the future. We thus created *UnChAnTies* our in-house clustering and pepMSA workflow and applied the formerly described approach to antigen peptides.

## III   METHODS

### A   Data mining

Over the last fifteen years, the field of immunology flourished in an unprecedented way, thanks to the rapid expansion of immunologic experimental techniques [3] and the development of a theoretical framework for the comprehension of the diversity in the immune system [12]. So far, despite many efforts [13], limitations in the direct access to molecular data persist, in particular, due to privacy issues. This situation strongly limits a wider comprehension of the CIC and other features of cancer. Fortunately, open-access resources are available. Our dataset is a subset of the IEDB [14], which collects $> 1.6$ million of experiments from $\approx 19500$ scientific publications. The total amount of epitopes now reaches. This effort sums up to detailed and standardized experimental data regarding more than $\approx 200000$ epitopes are now freely and easily accessible via browser. We searched for the following kinds of cancer, obtaining a *.fasta* file for each of them. We provide a summary of the dataset characteristics:

### B   Finding Clusters in the Immunopeptidome

Clustering is a central challenge in data mining and, due to their astonishing diversity, in molecular data as well. Typically, next-generation sequencing (NGS) produces a vast array of data that are counterintuitive to interpret if not grouped. This problem is a long-standing and well understood one in the context of proteins [15], [16], where a plethora of resources accumulated in time.

Initially, clustering proteins involves the analysis of protein families to determine unknown sequences. This process is, for instance, described in [15] which applies protein domains to obtain a clusterization of the datasets. In particular, this allows the assignment of functions to predicted proteins by comparison with proteins that belong to the same group, so characterized by joint annotations.

During our analysis, this process is not directly applicable. In fact, it is unknown how the peptides cluster, and it is consequently impossible to provide joint annotations involving this methodology.

To gain clustering information, it is critical underly that the grouping emerges because classes of proteins share some structure at the sequence level.

As explained in [17], it is possible to perform Multiple Sequence Alignment (MSA) to illustrate evolutionary related sequences. That accounts for evolutionary properties (eg, mutations, insertions, deletions, and rearrangements) and aligns the different sequences reflecting evolutionary traits. For instance, a method could involve adding gaps to the sequence: the additions represent insertions or deletions that occurred in the evolutionary path and allow the homologous positions to be aligned.

Different kinds of alignment may be possible. In particular, during this work, the researched alignment is functional. Furthermore, it is foreseen that, if the alignment is functional, the aligned positions are expected to support similar functions and so to be homologous. It is still important to consider that, even if it is reasonable to expect an overlap, the complexity of evolutionary changes does not allow a complete agreement.

But, if we assume that the sequence entirely determines function, then it is possible to perform a multiple sequence alignment, for which a vast literature exists [17].

In any case, it is important, at the conceptual level, to note that many of the MSA algorithms rely on the identification of motives, with which proteins recognize their interaction partners. Linear motives can be regarded as a continuous approximation of highly recurring subsequences, which is the central object of the *critical variable selection algorithm.* Thus, motives should be regarded as an ansatz for a more general structure. Once we build *.fasta* file for each cancer type, we grouped such files into a unique one, as if it comes out from a unique sampling process. We then choose, among the many algorithm available *Hammock* [18], which contemporary performs both clusterings (finding the correspondent of the protein families, which we will call *communities*) and performs multiple sequence alignment, the data structure to feed in CVS. We run *Hammock* with standard parameters, obtaining a total of $N_{Cluster} =$

We further checked that the quality of the obtained MSA reaches the standards.

|   |   | cov | pid | 1 [ . ] 15 |
|---|---|---|---|---|
| 1 | 1014_1 | 100.0% | 100.0% | -YADVDPENQNFLLE |
| 2 | 1014_2 | 92.9% | 100.0% | -YADVDPENQNFLL- |
| 3 | 1014_3 | 85.7% | 100.0% | -YADVDPENQNFL-- |
| 4 | 1014_4 | 78.6% | 100.0% | -YADVDPENQNF--- |
| 5 | 1014_5 | 64.3% | 55.6% | ----VRKDLQNFL-- |
| 6 | 1014_6 | 57.1% | 50.0% | ----VRKDLQNF--- |
| 7 | 1014_7 | 64.3% | 66.7% | ----SIPELQNFL-- |
| 8 | 1014_8 | 64.3% | 33.3% | ---SIKAQLQNF--- |
| 9 | 1014_9 | 64.3% | 33.3% | ---YQVGINQRF--- |
| 10 | 1014_10 | 64.3% | 33.3% | ---YQASYNQSF--- |
| 11 | 1014_11 | 71.4% | 20.0% | --TYSPALNKMF--- |
| 12 | 1014_12 | 71.4% | 30.0% | --LYQPNFNTNF--- |
| 13 | 1014_13 | 71.4% | 20.0% | --YYIQNGIQSF--- |
| 14 | 1014_14 | 78.6% | 27.3% | -TYFSAPGVMNF--- |
| 15 | 1014_15 | 78.6% | 18.2% | -AYYPAQGVQQF--- |

**Fig. 1:** An example of multiple sequence alignment from the retrieved peptides. ([**arwrf**])

## C  Critical Variable Selection

In contemporary molecular biology, objects are mainly investigated through their sequence. Such sequences are written with different (but related ) alphabets. The alphabet belonging to proteins is based on 21 amino acids. During 70's Anfinsen [19] postulated that, under very general conditions, the folding of a protein (and its function) should be completely determined by its amino acid sequence. Unfortunately, the mapping between sequence, folding, and function is a longstanding and unsolved problem. Several databases of homologous (ie performing the same function) proteins exists [11], [20], [21] and they were investigated in recent works based on model-oriented and model-free approaches [22], [9], [10]. One of the leading ideas in the field is that every algorithm that tries to do reverse-engineering the ensemble of proteins should be formulated as an optimization problem, which nature solves by sampling aminoacid sequences as configurations.

Similar efforts were still not applied in the context of peptides and, in particular, with antigens. If evolutionary forces select optimal peptides in cancer, we expect that they group according to some, still unknown, principles. In the former section, we described how we obtained the peptide clusters from a heterogeneous dataset, in the same way, homologous proteins are retrieved from a wide diversity of experiments. Applying a model-free method mutated from statistical physics [10] we aim to find if the internal space of sequences building up each cluster has an internal structure that can help us to shed light on their function. Once we obtained the MSA for each peptides cluster, we characterized its internal structure with critical variable selection. Albeit detrimental for the host, tumor cells pursuit their perpetration through the optimization of a set of tasks [23]. With this analogy in mind, we considered *CVS* as a suitable tool to map the immunopeptidome.

We can translate the multiple sequence alignment structure into

a matrix form. Let be

$$s_{i,j} \in \{A,B,C,...\} \quad i = 1...M, \ j = 1...L \qquad (1)$$

The $j$-th amino acid of the $i$-th sequence. We can introduce, for the single sequence, the notation $\vec{s}_i$.
This setting can be represented in a matrix form as follows.

$$\begin{bmatrix} m_{1,1} & m_{1,2} & \cdots & m_{1,C} \\ m_{2,1} & \ddots & & \\ \vdots & & & \\ m_{F,1} & & & m_{F,C} \end{bmatrix} = \begin{bmatrix} v_{1,1} \\ v_{2,1} \\ \vdots \\ v_{F,1} \end{bmatrix} \qquad (2)$$

The concept of *CVS* comes out by thinking of a sequence as the solution of an optimization problem with an unknown function and unobserved (latent variables). Following [10], the frequency with which a sub-sequence occurs in a given set of realizations is a reliable proxy for the underlying function being optimized. Once the sub-set of sites is fixed, sites harboring functional and structural information are very peaked in a frequency distribution, while highly non-conserved ones tend to have a flatter distribution. Thus, the heterogeneity in this distribution is a proxy for the relevance of the biological process of such a site. This observation imposes to look for subsets of variables such that the frequency with which the corresponding subsequences occur has a larger heterogeneity in the MSA.
Let us assume that the MSA spans $L$ sites and that among these only $l < L$ are relevant for the peptide to carry its function in the immunopeptidome. We introduce the index for the subset of such variables $I \subseteq \{1,2,..,L\}$ and we will call it slice, with meaning cleared in the following. We can thus think of each sequence as composed of relevant/non-relevant sites.
For example, let us consider $L = 8$ and $I = 1,4,5,7$. We can rearrange a set of sequences (here just one for sake of simplicity) as follows:

$$\vec{s}_i = (s_1, s_2, s_3, s_4, s_5, s_6, s_7, s_8) =$$
$$= (\underbrace{s_1, s_4, s_5, s_7}_{\text{relevant}} \underbrace{s_2, s_3, s_6, s_8}_{\text{non-relevant}}) = (\vec{s}_i^{\ \rho}, \vec{s}_i^{\ \overline{P}})$$

We can now define two information-theoretic measures regarding the sub-matrix of $S$ where we only consider the columns identified by $I_\rho$. We thus drop the index $\rho$, remarking that $I_\rho$ will be fixed until where the contrary is mentioned.

$$k_I(s) = \sum_{j=1}^{L} \delta(s, s_j) \qquad (3)$$

Is the number of times that a sub-sequence $s$ appears in the I-slice of the MSA.

$$m_I(s) = \sum_s \delta(k, k_I(s)) \qquad (4)$$

is the number of sub-sequences appearing exactly $k$ times. Extending the former example, we can visualize the relevant/non-relevant slicing procedure fig. 2 and how the practical calculation of $k_i(s)$ and $m_I(s)$ is performed.

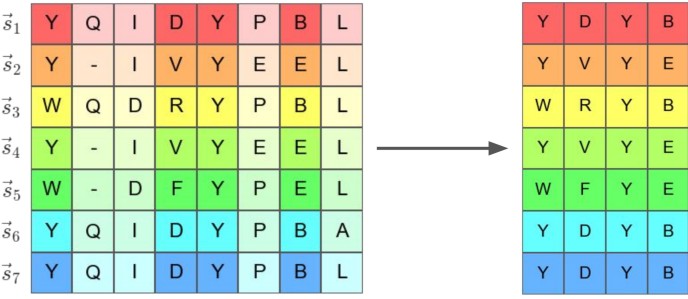

**Fig. 2:** Entropies are calculated one the subsequences sites $I$ is fixed. After slicing the MSA we obtain $H[k_I], H[s_I]$. Considering $I = \{1,3,4,5\}$ for this example, one get $k([YDYB]) = 3/7, k([YDYB]) = 3/7, k([WRYB]) = 1/7, k([WFYE]) = 1/7$. For the $m$s: $m(k = 3) = 1, m(k = 2) = 1, m(k = 1) = 2$

We measure the relevance of the sub-sequences generated by the slicing We say that a given slicing index $I$, unveils the relevance of the sites which is referred to with the following entropy, which is related [10] to the number of states that the slice I can distinguish using their sampling frequency:

$$H[k_I] = -\sum_s \frac{k m_I(k)}{M} \log_2 \frac{k m_I(k)}{M} \qquad (5)$$

When $H[k_I] > H[k_{I'}]$ it means that the slice $I'$ produces a broader distribution of frequencies meaning that some structure with a strong biological meaning emerges considering $I'$ as the slice of the relevant sites. Thus, for fixed slice length $|I| = n$ we want to find the slice that maximises the relevance:

$$\tilde{I}(n) = \arg \max_{I:|I|=n} H[k_I] \qquad (6)$$

The description of the algorithm with which we find such $I$ is demanded in the next section. Finally, we note that $H[k_I]$ is different from the entropy of the sequence

$$H[s_I] = -\sum_s \frac{k_I(s)}{M} \log_2 \frac{k_I(s))}{M} = -\sum_k \frac{k m_I(k)}{M} \log_2 \frac{k_I(s)}{M} \qquad (7)$$

Which is anyway related to $H[k_I]$ by $H[k_I] = H[s_I] - \sum_k \frac{k m_k}{M} \log m_k$ from which we obtain the bound $H[k_I] \leq H[s_I]$

Here, starting with the same MSA a different choice of the slicing set $I$ at a fixed number of relevant sites $n$ can result in a different outcome in the histogram of $k$. This histogram is the central object of our investigation since it defines both $H[k_I]$ and $H[s_I]$.

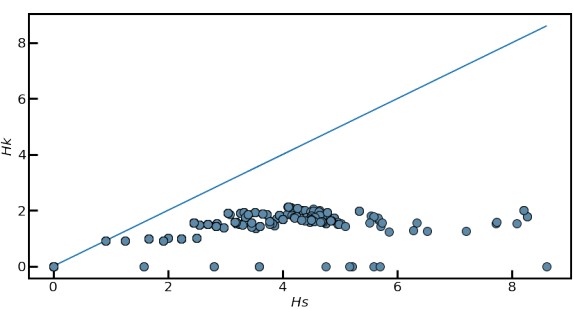

**Fig. 3:** Each cluster has highly diverse entropy landscape, which, in any case, respect the bound $H[k_I] \leq H[s_I]$.

### D  Algorithm Implementation

Due to the specificity of the *CVS* framework, we follow [24] in defining proposing an ansatz for finding $\tilde{I}(n)$. The idea is to go towards the global optimum of $H[k_I]$ with a gradient ascent algorithm, with a Monte-Carlo-inspired implementation. This algorithm was then performed on every cluster of the dataset, to obtain a characterization of the internal variables involved in the ligand-receptor recognition between T-cells receptors and cancer surface proteins, namely the antigen.

The idea is to explore as much as possible all local maxima of the entropy landscape. To do so we set several iterations $T$ that is the maximum number of states that can be explored for each trajectory. Inside each trajectory, a new state is proposed employing a *rejection-acceptance rule*: if the entropy of the new state is higher than the old one, then the proposed state is accepted as a new one. The new state is generated randomly choosing, one of the sites in $I$ and changing it with a random one. We call this rule *flip Algorithm 1*. To ensure convergence, we set a maximum number of rejections *rejections*. This part of the algorithm is supposed to reach, at best, a local maximum. To ensure a complete exploration of the entropy landscape we run this algorithm $R$ times with a randomly initialized state.

This algorithm runs over all the MSA that characterize each cluster, looking for different large-scale correlations.

## IV  RESULTS

Now, we can consider the outcome of our *critical variable selection* procedure. We can introduce the following measure of relevance for a single site: let $c_i$ be the number of times (eventually normalized) that a site appears as an argument of $I_n$ when

---

**Algorithm 1** Gradient-Ascent Relevance optimization

```
1:  for t in R do                      ▷ Random initialization
2:      for t in T do                  ▷ Monte Carlo
3:          while p ≤ rejection do
4:              I' ← flip(I)
5:          end while
6:          if H[k_I] ≤ H[k'_I] then
7:              I ← I'
8:          else
9:              p+ = 1
10:         end if
11:     end for
12: end for
```

it reaches a local maximum. As we see from 4, this measure exhibits, for all clusters a universal curve characterized by a peak. Such a pattern suggests that there exists a common organizational principle of the immunopeptidome. This should not be surprising, since every sequence in such an ensemble participates in the ligandome, and perform the same function. In any field of sciences, it is central in the comprehension of a phenomenon to identify if a pattern can be reproduced by a random model. If the random model produces a pattern comparable to the data, then one can state that variability in data can be described as a deviation from a random model. Now, a typical method to test this idea [25] is to produce new data from the available ones. We implemented such a procedure as follows. We substitute every column with a similar one but randomly permuted. We do this random permutation independently for each column, to destroy the identity of sites. Doing the same along rows we destroy the identity of sequences as well. While this procedure scrambles both samples and variables, it does not modify the entries of the data matrix.

Thus, we obtained a randomized version of the data against which we could compare the empirical ones. If after randomization the pattern would look the same, that would mean that even original data were random. After the randomization procedure, we compared the two $c_i$ patterns finding the second one. exhibit a $c_i$ that is constant plus noise, meaning that the universal peak we find in data is a strong signal of the spatial structure that relevance assumes in the immunopeptidome.

## V  CONCLUSION

During the present work, we explored and characterized the diversity between antigens. It has been shown that critical variable selections can identify the original function of the antigen.

What has been additionally shown, is that the sites have a universal organization. That may be caused by the fact that all sequences we are considering belong to surface proteins, so they

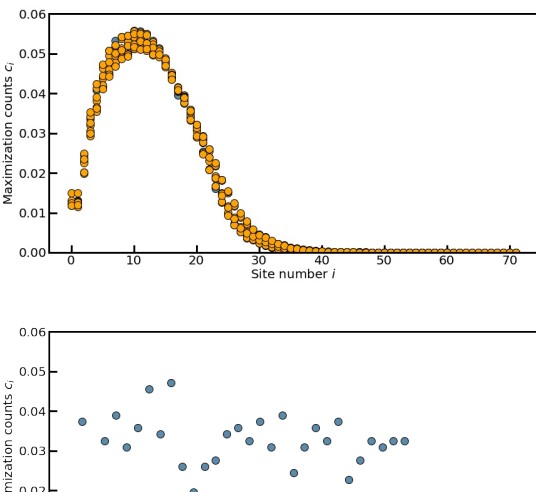

**Fig. 4:** Each cluster exhibits a universal pattern in the organisation of relevance along the biological sequence which is strongly non-random. In fact, random data present flat distribution, as shown in the second picture, contrary to the relevant shape of real data (first picture) which is modal.

are part of the ligandome. As described by [26], despite molecular diversity, universal laws that regulate universal ligandome interactions hold. In future work, it will be crucial to collect a larger amount of data, to let statistical differences emerge more clearly. In the context of random models, [27], deviation of the data from a random baseline is a valuable direction for further investigations.

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
