# OpenReview forum: "Immunopeptidome Cluster Profiling with Critical Variable Selection"
_uoft.ai/University_of_Toronto/2021/ProjectX — Submitted to ProjectX2021_

### Official Review · Reviewer_p9QL · 2022-02-12
**Immunopeptidome Cluster Profiling with Critical Variable Selection**

**Rating:** 4
**Confidence:** 4

**Review:**

Major strengths
-Borrowing the successful idea from statistical physics and successfully transforming the cluster-specific feature selection as an optimization problem to find related features that characterize the clusters or communities of antigens.

Major limitations
-	The manuscript writing can be significantly improved. It looks the authors did not read it after the draft was completed. This is unacceptable.  For example, it looks like Figures 1,3, and 4 have never been cited in the manuscript. there are too many small paragraphs, which can be merged. Furthermore, it looks some contents are missing as shown here “. We provide a summary of the dataset characteristics:” since I did not see the summary in the manuscript.
-	The CVS method or framework should be benchmarked with other existing methods.
-	The study looks to have not been completed yet since I did not see any results about the cluster structures and the identified features in the result section although a large data set was mentioned in Section III.A.
-	It is not a good strategy to select features using CVS after the clustering is done. In this computational pipeline, it will be great if the feature selection is performed before the clustering step.
-	Overall, the paper is very hard to follow.

---

### Official Review · Reviewer_pBUi · 2022-02-13
**Immunopeptidome Cluster Profiling with Critical Variable Selection**

**Rating:** 6
**Confidence:** 4

**Review:**

Immunopeptidome Cluster Profiling with Critical Variable Selection

The authors have proposed UnChAnTies (Unsupervised characterization of antigen communities) to detect biological motifs in cancer antigen peptides. Considering this is a work done in 3-5 months and only involves undergraduate students, I am very impressed with the quality of work. Generally the paper is well written, decent literature review, and a good solution for the problem. However, I will review and criticize this work as I do for any high tier research conference.

Major comments:

1. The authors need to compare their work with the SOTA (state of the art) methods in the field. The result section is very short and very few experiments have been done to convince the reader that the UnChAnTies works and provides reasonable results and UnChAnTies is better than existing methods.

2. UnChAnTies code is not available to me to run for toy examples and/or replicate the results.

3. The main text needs polishing and re-writing. Some paragraph are missing:
Page1: “Human Leukocyte Antigen (HLA) contributes to more than half of the four to five million single-nucleotide polymorphisms (SNPs) in each genome.” is wrong
Page2: “We provide a summary of the dataset characteristics:” The rest of the text is missing
Page3: “ We run Hammock with standard parameters, obtaining a total of NCluster =” The rest of the text is missing
“Motives”, I assume you mean motifs!
Page3, Figure 1. I think [arwrf] is a citation from latex!
Page3, Figure 1. Caption needs to explain the example in more detail. What is vod and pid for example.
Page4, Figure 2. I think that the provided example has a problem.

Connection to Current Science (science and practice)
Reviewer: I will give a score of 1.5. I think that the authors did a good job explaining the motivation of their work. Regarding adding new to the field, I am not expert to make this judgment but

Clarity of Communication
Reviewer: I will give a score of 1. The paper is well written but some sections were hard to follow.

Methodological Quality
Reviewer:  I will give a score of  2. I think the provided method is a simple method. However, I am not sure the method works in practice.

Reproducibility
Reviewer: I will give a score of 1. No code was provided to reproduce the results.

---

### Official Review · Reviewer_Cu8m · 2022-02-15
**The authors propose the use of a critical variable selection approach to identify relevant peptide multiple sequence alignment defined by clustering a custom dataset of epitopes (immunopeptidome) with the hopes of this approach enables researchers to identify targets for therapeutic interventions.**

**Rating:** 5
**Confidence:** 3

**Review:**

Overall, I commend the authors for their efforts and innovative thinking on this timely topic. The manuscript is well organized and follows a clear structure and logic, however, I have some comments regarding the quality, clarity, and significance of this work.

**Major comments**

The authors may be making strong implicit assumptions by pulling all fasta files within a cancer type without more careful considerations. For instance, how can the authors ensure that the resulting clusters are not artifacts or correlate with well-stablished clinical characteristics?

Why picking the Hammock algorithm over others? This is unclear from the description in section III B. Is this because it can perform both clustering and multiple sequence alignment?

There are a few hyper-parameters in the algorithm implementation in section III D, e.g. R, T, rejections. It is unclear how the authors decide on specifying those values. How can they guarantee that their choice was appropriate?

There is no clear path to implementation/translation. How will other researchers be able to interpret data resulting from this approach? To me this the most critical aspect of the manuscript as the authors made no attempt to interpret the data. What does it mean that you see a peak (maximum)?

**Minor comments**

I find some sections with incomplete ideas or lack of supporting tables/figures, e.g. end of section III A, end paragraph of III B

As a consequence of the previous point, there in an incomplete characterization of the data used in this manuscript.
Unclear notation for elements in equation (2), what are m, v, elements and F, C dimensions? Unclear how these relate to equation (1).


**Pros**

Mined a large dataset of open-access experimental data.
Used an unsupervised approach to cluster sequences focusing on functional alignment.
Use a model-free approach (critical variable selection) to identify potentially relevant structures (antigen communities).

**Cons**

The manuscript lacked any attempts to validate the findings beyond the described randomization procedure.
No code was made available to reproduce the results.
The evaluation metrics did not consider any uncertainty associated to them; how can the authors evaluate for potential false positives?

---

### Official Review · Reviewer_uKGG · 2022-02-16
**This study aims to identify in an unsupervised manner characteristic sequence features of the immunopeptidome using "critical viable selection". While the problem and method are interesting, the paper makes incorrect biological assumptions (which oddly may not affect the application as much) and the results and conclusions are very rudimentary.**

**Rating:** 4
**Confidence:** 4

**Review:**

This study aims to identify in an unsupervised manner characteristic sequence features of the immunopeptidome using "critical viable selection". The basic idea is to group peptides identified as bound to MHC I by immunoproteomics through multiple sequence alignment and then apply "critical variable selection" to identify characteristic features of each MSA/cluster. The authors report that every cluster has a shared non-random structure (by comparing to permuted data).

Figuring out the features of the immunopeptidome is indeed important, but unfortunately, the authors seem a bit mix on what the immunopeptidome *is*. An immunopeptidomics experiment usually only measures if a peptide is bound to MHC (and in these data, this would be MHC I). This is affected by the peptide and by the MHC allele, so each immunopeptidomic experiment is usually interpreted in the context of that allele. Alas, to my best understanding of the writing, the authors confuse this with recognition by the T cell receptor (TCR), which is a ternary interaction MHC-peptide-TCR, and I do not believe has been captured by the data they use. Moreover, the authors focus on "selection" in the context of neoantigens is erroneous. The vast majority of neoantigens will be due to passenger (non selected) mutations and at most should be selected against (negatively) not for (positively) due to the selective pressure exerted by the immune system (which would in turn lead to immune editing, etc). Thus, large swaths of the text in the paper are just incorrect.

Next, the authors take two computational steps. They group sequences by a combination of MSA and clustering, If I followed the MSA descriptions, they allow gaps a major issue for any later interpretation (as these are short presented peptides!). Finally, they apply CVS; I thought that was a potentially interesting idea (if applied to the right input in the right way), but there is simply no validation or interpretation that allows us to evaluate this.

Thus, while the problem is surely important and the core method may be interesting, the study needs to be corrected around assumptions, pre processing and interpretation to demonstrate the utility of the approach.

---

### Decision · Program_Chairs · 2022-02-19

NA